# Dietary Protein Intake during Pregnancy Is Not Associated with Offspring Insulin Sensitivity during the First Two Years of Life

**DOI:** 10.3390/nu12051338

**Published:** 2020-05-08

**Authors:** Brittany R. Allman, D. Keith Williams, Elisabet Børsheim, Aline Andres

**Affiliations:** 1Arkansas Children’s Nutrition Center, Little Rock, AR 72202, USA; ballman@uams.edu (B.R.A.); williamsdavidk@uams.edu (D.K.W.); 2Arkansas Children’s Research Institute, Little Rock, AR 72202, USA; 3Department of Pediatrics, University of Arkansas for Medical Sciences, Little Rock, AR 72202, USA; 4Department of Biostatistics, University of Arkansas for Medical Sciences, Little Rock, AR 72205, USA; 5Department of Geriatrics, University of Arkansas for Medical Sciences, Little Rock, AR 72205, USA

**Keywords:** pregnancy, offspring, protein, insulin resistance, insulin sensitivity, HOMA-IR, obesity, plant protein, glucose, insulin

## Abstract

Literature describing a relationship between dietary protein intake during pregnancy and offspring insulin resistance are equivocal perhaps because of the lapse between maternal and offspring measurements (~9–40 years). Thus, we evaluated protein intake in healthy women [*n* = 182, mean ± SD; body mass index (BMI): 26.2 ± 4.2 kg/m^2^] in early pregnancy (8.4 ± 1.6 weeks, EP), late pregnancy (30.1 ± 0.4 weeks, LP), and averaged throughout pregnancy, and determined the relationship between protein intake and offspring homeostatic model assessment of insulin resistance (HOMA2-IR) at 12 (12mo) and 24 (24mo) months. EP protein (g·kg^−1^·day^−1^) did not associate with HOMA2-IR at 12mo (β = 0.153, *p* = 0.429) or 24mo (β = −0.349, *p* = 0.098). LP protein did not associate with HOMA2-IR at 12mo (β = 0.023, *p* = 0.916) or 24mo (β = −0.442, *p* = 0.085). Finally, average protein did not associate with HOMA2-IR at 12mo (β = 0.711, *p* = 0.05) or 24mo (β = −0.445, *p* = 0.294). Results remained unchanged after adjusting for plant protein intake quartiles during pregnancy, maternal BMI, and offspring sex and body fat percentage. Additionally, these relationships did not change after quartile analysis of average protein intake, even after considering offspring fasting time and HOMA2-IR outliers, and maternal under-reporters of energy intake. Protein intake during pregnancy is not associated with indirect measurements of insulin sensitivity in offspring during the first two years of life.

## 1. Introduction

We have recently shown that there is a positive relationship between dietary protein intake during pregnancy and maternal insulin sensitivity estimated using the metabolic clearance rate of glucose after an oral glucose tolerance test administered in late pregnancy (~30 weeks) [1]. However, these relationships may have been mediated by obesity status [1], evidenced by the weakening or disappearance of these relationships after accounting for BMI. Although there is a growing understanding of the relationship between protein intake during pregnancy and maternal insulin resistance, very little is known about the relationship between protein intake during pregnancy and offspring insulin sensitivity [2,3,4,5]. These relationships are important to elucidate considering the potent influence of the maternal environment on offspring health. In particular, advocacy for improved nutrition during the first 1000 days of life (which includes the entirety of the gestational period) to improve offspring development is increasing [6]. This increasing support is rooted in the Developmental Origins Hypothesis which proposes that stressors during pregnancy provoke metabolic adaptive changes in the growing fetus both acutely and chronically [7].

Nonetheless, the current literature regarding the association between maternal protein intake and offspring insulin sensitivity is sparse and mixed [2,3,5]. One report [5] noted that there was a relationship between maternal protein intake at 30 weeks gestation (≤0.96 (Q1), 0.97–1.13 (Q2), 1.13–1.29 (Q3), and >1.29 g·kg^−1^·day^−1^ (Q4), *p* = 0.007) and offspring insulin increment in response to a 75 g glucose load [(insulin concentration at 30 min - fasting insulin concentration)/(glucose concentration at 30 min), used as a measure of insulin secretion [8,9] at age 40 years old. Conversely, another report [3] noted no association with fasting plasma insulin or HOMA-IR in offspring aged 9–16 years old, at similar levels of protein intake in the maternal diet at 25 weeks gestation [0.94–1.08 (Q1), 1.17–1.22 (Q2), 1.28–1.34 (Q3), 1.41–1.57 g·kg^−1^·day^−1^ (Q4)].

Furthermore, taking the type of dietary protein into consideration, one study [4] reported no correlations between fasting blood glucose or insulin measures in male or female offspring at age 20 years old and maternal animal or vegetable protein intake at gestational week 30. However, the specific type of animal protein may be important to consider because another report [3] noted a positive relationship between red and processed meat intake during gestational week 25 and HOMA-IR and fasting insulin of offspring (aged 9–16 years) born to control women (healthy pregnant women) and male offspring of women with GDM [3] after adjusting for parental sociodemographic position, maternal age, parity, pre-pregnancy BMI, smoking, energy intake, and offspring age and sex. White meat and dairy sources of animal proteins were not related to offspring HOMA-IR or fasting insulin concentrations, however.

There are a few limitations to the current literature. The described studies only assessed dietary intake in late pregnancy (25–30 weeks of gestation) and are thus not reflective of any potential change in dietary protein intake throughout gestation. Although some evidence points to a limited change in dietary patterns throughout pregnancy [10], it is known that dietary protein requirements increase significantly from early to late pregnancy (early pregnancy: 11–20 weeks, 1.2 g·kg^−1^·day^−1^; late pregnancy: 30–38 weeks, 1.52 g·kg^−1^·day^−1^) [11]. Further, the offspring were not assessed until much later in life (9–40 years old) [2,3,4,5], and thus, there is a paucity of data assessing offspring in early life (e.g., infants, toddlers). The offspring are most impressionable in this early time period because of the temporal proximity to pregnancy and because of continued direct influence from the mother through interactions such as lactation.

Overall, research is required to analyze the impact of protein intake during pregnancy on offspring insulin sensitivity in early life. Therefore, the aim of this study was to assess the association between protein intake during pregnancy and offspring insulin sensitivity during the first two years of life. We hypothesized that there will be no relationship between protein intake during pregnancy and offspring insulin sensitivity measures, and if there is a positive relationship, it will be driven by maternal obesity status and the type of protein (e.g., plant) consumed.

## 2. Materials and Methods

This study took advantage of a clinical dataset from the *Glowing* cohort (ClinicalTrials.gov identifier: NCT01131117) at the Arkansas Children’s Nutrition Center. *Glowing* was established to describe the maternal programming of offspring obesity in pregnant women of various BMI classifications. This study assessed a sub-group of the original cohort because complete analysis required: (1) completion of at least one of the two maternal visits, (2) completion of at least one of the two offspring visits, (3) complete and available dietary data, (4) successful offspring blood draws, and (5) successful offspring body composition scans. All experimental procedures were conducted according to the Declaration of Helsinski principles and were approved by the Institutional Review Board at the University of Arkansas for Medical Sciences (Ethics Approval Number: 110889, Date of Approval: 28 July 2009). Written informed consent was provided by each participant.

Participants were recruited early in pregnancy (<10 week) and were included if they satisfied the following criteria: second parity, singleton pregnancy, ≥21 years old, and if they conceived without the use of fertility treatments. Participants were excluded if they satisfied any of the following criteria: preexisting medical conditions (e.g., gestational diabetes mellitus, chronic renal failure, hypertension, malignancies, seizure disorder, lupus, drug or alcohol use, serious psychiatric disorders), sexually transmitted diseases, medical complications during pregnancy (e.g., gestational diabetes mellitus, pre-eclampsia), took medications during pregnancy known to influence fetal growth (e.g., thyroid hormone, glucocorticoids, insulin, oral hypoglycemic agents), and if they performed excessive physical activity (an athlete engaged in a professional sports activity). Offspring were included if they met the following criteria: healthy, normal infants at birth, successful, and full-term pregnancy (≥37 weeks gestation). Offspring were excluded if they met the following criteria: took medications known to influence growth and development, or had medical conditions known to influence growth and development. Mothers were classified based on BMI: normal weight (18.5–24.9 kg/m^2^, NW; *n* = 82, 45%), overweight (25.0–29.9 kg/m^2^, OW; *n* = 65, 36%), or obese (>30 kg/m^2^, OB, *n* = 35, 19%).

To limit excessive gestational weight gain (GWG) during pregnancy (excessive GWG at 30 weeks: >12.0 kg for normal weight, >8.6 kg for overweight, and >6.8 kg for obese women, Institute of Medicine recommendations for GWG [12]), trained nutritionist provided coaching of nationally-recognized nutrition and GWG guidelines [13] to the future mothers [14]. Reported results are from early pregnancy (EP, mean ± SD, 8.4 ± 1.6 weeks), late pregnancy (LP, 30.1 ± 0.4 weeks), offspring aged 12 months (12mo, 11.7 ± 1.8 months), and offspring aged 24 months (24mo, 22.2 ± 6.4 months). A medical history questionnaire was administered at EP and was updated at each subsequent visit.

### 2.1. Anthropometrics and Body Composition

For the mother, weight was measured to the nearest 0.1 kg using a standing calibrated digital scale (Perspective Enterprises, Portage, MI, USA) and height was measured to the nearest 0.1 cm standing against a wall-mounted stadiometer (Tanita Corp., Tokyo, Japan). For the offspring, weight was measured to the nearest 0.01 kg using a tared scale (SECA 727; SECA Corp, Ontario, Canada) while the infant was only wearing a clean diaper. Length was measured to the nearest 0.1 cm using a length board with a fixed headpiece and sliding foot piece (Easy Glide Bearing Infantometer; Perspective Enterprises, Portage, MI). Body composition of the offspring was assessed using nuclear magnetic resonance (NMR; EchoMRI-AH, Echo Medical Systems, Houston, TX, USA).

### 2.2. Insulin Resistance Analysis

Mothers were asked to visit the laboratory when their child was approximately 4-h fasted at the 12mo time point and approximately 12-h fasted overnight at the 24 mo time point. At the time of their visit to the laboratory, the mother was asked about the time when the child last ate. Blood samples were collected from the offspring after 2 ± 2 (average ± SD) hours of fasting at 12mo (range: 0–15 h), and 9 ± 6 h of fasting at 24 mo (range: 0–17 h). Samples were collected in EDTA-coated plasma vacutainers (Becton, Dickinson & Company, Franklin Lakes, NJ, USA) for analysis of insulin (Mesoscale Discovery Platform Multi-Array Assay System, Gaithersburg, MD, USA), and serum vacutainers (Becton, Dickinson & Company, Franklin Lakes, NJ, USA) for analysis of glucose (Randox Daytona Clinical Analyzer, Randox Daytona, Crumlin, UK). Plasma concentrations were measured in duplicate using commercially available enzyme linked immunosorbent assays kits, according to the manufacturer’s instructions. The Oxford Centre for Diabetes, Endocrinology and Metabolism homeostasis model assessment-2 calculator (University of Oxford, Oxford, UK, https://www.dtu.ox.ac.uk/homacalculator/) was used to estimate HOMA2-IR and HOMAβ (indicator of insulin-secreting ability) from fasting plasma glucose and insulin concentrations [15].

### 2.3. Dietary Intake

Dietary intake was assessed at EP and LP, using 3-day food records (two week days and one weekend day) with the Nutrition Data System for Research (NDSR, Nutrition Coordinating Center, University of Minnesota, MN) software. Dietary intake was also averaged between the two time points. Dietary intake of total protein (TP), animal protein (AP), and plant protein (PP) relative to body mass (kg) were estimated.

### 2.4. Statistical Analysis

Power analyses was estimated using a classic approach (Maslova et al., 2017 [3]) for multiple regression analyses in the presence of seven covariates. The current sample size (*n* = 182) had at least 80% power to detect a ‘small’ Cohen effect size of 0.05 at the significance level of 5%. Significance for all tests was set to *p* < 0.05. Data are presented as mean ± SD.

#### 2.4.1. Linear Model of Maternal Protein Intake vs. Offspring HOMA2-IR

Two linear models with offspring HOMA2-IR as the response variable were fitted to estimate: (1) the unadjusted relationship of dietary protein intake during pregnancy (independent variable) to the dependent variable and (2) the relationship of dietary protein intake during pregnancy (independent variable) to dependent variable adjusted for dietary plant protein intake quartiles during pregnancy and maternal BMI at the respective gestational week (EP, LP), and offspring body fat percentage at the respective age (12mo, 24mo). Plant protein quartiles were dependent on the time point of interest [EP: 0.12–0.28 (Q1), 0.29–0.35 (Q2), 0.36–0.42 (Q3), 0.43–0.66 (Q4); LP: 0.16–0.30 (Q1), 0.31–0.36 (Q2), 0.37–0.41 (Q3), 0.42–0.84 (Q4); average between EP and LP: 0.13–0.30 (Q1), 0.31–0.35 (Q2), 0.36–0.40 (Q3), 0.41–0.63 (Q4)].

#### 2.4.2. Quartile Analysis of Average Maternal Protein Intake vs. Offspring HOMA2-IR

These relationships were also analyzed using quartile analysis of average protein intake between EP and LP (Q1: 0.47–0.72; Q2: 0.72–0.88; Q3: 0.88–1.0; Q4: 1.0–1.5 g·kg^−1^·day^−1^) compared to offspring HOMA2-IR and HOMAβ at 12mo and 24mo, adjusted for offspring sex, as well as dietary plant protein intake quartiles, maternal BMI, and offspring body fat percentage at the respective time points (EP, LP, 12mo, 24mo). Because HOMA measures are often not normally distributed [16], we also conducted these analyses after logarithmic transformation.

#### 2.4.3. Posthoc Analyses of Quartile Analysis

Three separate posthoc analyses of the quartile analysis were conducted by: (1) adding offspring fasted time [time of blood draw – time last eaten (min)] to determine if fasted time impacted the model; (2) excluding maternal dietary underreporters of energy intake (energy intake:basal metabolic rate < 1.35) to determine if underreporting of energy intake impacted the model; and (3) using a robust regression utilizing the median (instead of the mean) to control for the effect of outliers of offspring HOMA2-IR.

## 3. Results

### 3.1. Participant Characteristics

Three hundred participants were recruited for the *Glowing* study. Many participants from the parent study were excluded for not satisfying the requirements for inclusion in the current analyses (i.e., a maternal dietary record from at least one of the two pregnancy time points; successful offspring blood draw during at least one of the two offspring time points). Specifically, of the total cohort (*n* = 182), 176 women completed a dietary food record at EP (8.4 ± 1.6 weeks), 176 women completed a dietary food record at LP (30.1 ± 0.4 weeks), and 150 offspring had successful blood draws at 12mo, and 133 offspring had successful blood draws at 24 mo. One woman at 30 weeks was missing BMI data, 24 offspring at 12mo and 40 offspring at 24mo were missing blood data, and 31 offspring at 12mo and 43 offspring at 24mo were missing body composition (body fat percentage) data and were therefore excluded from adjustment analysis. Table 1 shows the descriptive characteristics of the participants at baseline. Offspring insulin resistance, represented by HOMA2-IR did not differ between 12mo and 24mo (*p* = 0.999). Just over 50% of the pregnant women in our cohort gained over the IOM recommendation for GWG (not pictured in Table 1).

Table 2 shows dietary protein intake data at EP, LP, and an average between EP and LP. There were no vegetarians or vegans in our population of pregnant women. Pregnant women consumed more total energy (kcal/day) at EP compared to LP. In addition, they consumed less total protein relative to body mass, animal protein relative to body mass, and plant protein relative to body mass at LP compared to EP. Participants consumed more absolute plant protein at LP compared to EP.

### 3.2. Results of Linear Model of Maternal Protein Intake vs. Offspring HOMA2-IR

Multiple linear regressions were performed to predict offspring HOMA2-IR at 12mo and 24mo based on maternal total dietary protein intake at EP and LP (Table 3), as well as throughout pregnancy (average between EP and LP, not shown). The unadjusted model showed no relationship between: total dietary protein intake during pregnancy at EP and offspring HOMA2-IR at 12mo (β = 0.153, *p* = 0.429), total dietary protein intake during pregnancy at EP and offspring HOMA2-IR at 24mo (β = −0.349, *p* = 0.098), total dietary protein intake during pregnancy at LP and offspring HOMA2-IR at 12mo (β = 0.023, *p* = 0.916), or total dietary protein intake during pregnancy at LP and offspring HOMA2-IR at 24mo (β = −0.442, *p* = 0.0851). There was no relationship between average dietary protein intake throughout pregnancy and offspring HOMA2-IR at 12mo (β = 0.711, *n* = 123, *p* = 0.05), or between average dietary protein intake throughout pregnancy and offspring HOMA2-IR at 24mo (β = −0.445, *n* = 110, *p* = 0.294). After adjusting for dietary plant protein intake quartiles and BMI during pregnancy at the respective gestational week (EP, LP, average), offspring sex, and offspring body fat percentage at the respective age (12mo, 24mo), there were no significant findings (Table 3). Further, there was no relationship between average maternal protein intake and HOMA-β at either time point (12mo: β = 57.6, *p* = 0.08; 12mo: β = 1.29, *p* = 0.94). After logarithmic transformation of HOMA2-IR and HOMA-β, there were still no notable relationships between average maternal protein intake and offspring HOMA2-IR at either time point (12mo: β = 0.69, *p* = 0.06; 24mo: β = −0.63, *p* = 0.11), or HOMA-β at either time point (12mo: β = 0.47, *p* = 0.08; 24mo: β = 0.03, *p* = 0.85).

### 3.3. Results of Quartile Analysis of Average Maternal Protein Intake vs. Offspring HOMA2-IR

To best mirror the statistical approaches used in the discussed literature [3,5], quartile analysis was conducted to compare average protein intake between EP and LP (Q1: 0.47–0.72; Q2: 0.72–0.88; Q3: 0.88–1.0; Q4: 1.0–1.5 g·kg^−1^·day^−1^) and offspring HOMA2-IR at 12mo and 24mo, adjusted for offspring sex, as well as dietary plant protein intake quartiles, maternal BMI, and offspring body fat percentage at the respective time points (EP, LP, 12mo, 24mo). There were no significant relationships between the protein quartiles and HOMA2-IR values at any time point.

### 3.4. Results of Posthoc Analyses of Quartile Analysis

After adding offspring fasted time [time of blood draw – time last eaten (min)] to the model, the relationship between maternal protein intake and offspring HOMA2-IR at the respective time points did not change. Additionally, fasted time was not related to HOMA2-IR at 12mo (β = −0.000704, *p* = 0.19), but was related to HOMA2-IR at 24mo (β = −0.00150, *p* < 0.001).

Further, after excluding maternal dietary under-reporters of energy intake (energy intake:basal metabolic rate < 1.35) from this model, we found no significant difference in the findings (12mo vs. maternal average protein intake: *n* = 62, *p* = 0.874; 24mo vs. maternal average protein intake: *n* = 60, *p* = 0.327).

Lastly, to rule out an effect of HOMA2-IR outliers, post-hoc analysis was conducted using a robust regression utilizing the Huber weight function. Results revealed that there were no differences between offspring insulin resistance and maternal protein intake at respective time points (12mo vs. maternal average protein intake: *p* = 0.692; 24mo vs. maternal average protein intake: *p* = −0.776).

## 4. Discussion

With limited human research data available [3,4,5], the association between protein intake during pregnancy and offspring insulin sensitivity is unclear. Current evidence is limited by the large gap between dietary assessment during pregnancy and offspring insulin resistance measurements (~9–40 years). Therefore, for the first time, this study aimed to evaluate offspring insulin sensitivity in early life (12mo, 24 mo). Nevertheless, linear and quartile analysis of our data revealed that there were no associations between maternal protein intake at any time point (EP, LP, average) and offspring insulin resistance (12mo, 24mo). Our data showed that there was a substantial range in the percentage of daily energy intake contributed by protein at both time points (EP: 8.7%–25.4%; LP: 9.3%–28.4%), which is a strength of the current study.

Despite the numerous benefits of habitually consuming a diet higher in protein (e.g., increased satiety [17], increased energy expenditure and thermogenesis [18], weight loss/regulation [19]), an ongoing debate exists regarding the association between eating a diet high in dietary protein intake and metabolic health. For example, high dietary protein intake during pregnancy is related to both maternal [1] and offspring [5] insulin resistance. However, not only do these relationships weaken or disappear when accounting for early pregnancy BMI [1], but there are also some studies that refute these findings all together, clearly noting no association between dietary protein intake during pregnancy and offspring measures of insulin resistance [3,4]. In the present study, after adjusting for offspring body fat percentage (12mo, 24mo) and maternal BMI (EP, LP) at the respective time points (in addition to offspring sex and plant protein intake quartiles during pregnancy), results remained unchanged. Therefore, it seems that neither maternal BMI nor offspring body fat percentage directly affect the relationship between protein intake during pregnancy and offspring insulin sensitivity.

Our report of no relationship between a relatively wide range of maternal protein intake (LP range: 0.4–1.8 g·kg^−1^·day^−1^) and offspring insulin resistance measures is confirmed by another report in pregnant women in gestational week 25 that reported no relationship between maternal protein intake (1.04–1.75 g·kg^−1^·day^−1^, calculated from estimated body mass) and fasting plasma insulin or HOMA-IR in offspring aged 9–16 years old [3]. Considering other reports of a positive association between protein intake and insulin resistance measures (reviewed in Rietman et al., 2014 [20]), it is interesting to note that the protein intake in both studies spanned higher than protein requirements recently reported in late pregnancy (1.55 g·kg^−1^·day^−1^ [11]), yet neither were associated with offspring insulin resistance.

Although the hyperinsulinemic-euglycemic clamp is considered the gold standard for insulin resistance assessment [21], it poses serious ethical concerns in a toddler population (e.g., IV infusion, long commitment to laboratory) and has only been used in infant populations simultaneously undergoing critical care (e.g., preterm infants [22]). Furthermore, although other options to measure insulin resistance (e.g., glucose tests, modeling) have been used in older pediatric populations, toddlers are an overall understudied population [23]. Further, because our large cohort study drew one-time fasted blood samples at 12mo and 24mo, we were limited by which insulin resistance assessment we could potentially use. Because: (1) HOMA-IR was used in other work comparing the same variables [3]; (2) HOMA2-IR is a computerized updated model of HOMA-IR accounting for variations in hepatic and peripheral glucose resistance; (3) it has been used in a pediatric population [24]; and (4) our previous publication that serves as the precursor of this study also used HOMA2-IR [1]; and (5) the measurement takes both fasting glucose and insulin into account, we decided to use HOMA2-IR as an assessment of insulin resistance. Nevertheless, there was a significant spread in HOMA2-IR of the offspring at both 12mo (0.06–6.85) and 24mo (0.07–5.26). Importantly, the large variation between offspring fasting duration before the blood draw occurred (12mo: 0–15 h; 24mo: 0–17 h) did not affect the model and thus the conclusions, as revealed in the posthoc analysis. However, because the prevalence and the specific definition of insulin resistance using HOMA2-IR in toddlers (12–24 mo) has not been defined, it may be difficult to discover a potential relationship at this age. Therefore, future research should aim to define insulin resistance in the early years. Nonetheless, our posthoc analysis using a robust regression utilizing the median (instead of the mean, to determine if outliers were driving the model) determined that there was no change in the model, and therefore, an effect of particularly high HOMA2-IR values on our findings is likely non-existent.

Our data may have been limited because we did not perform measurements (e.g., urinary nitrogen excretion [25]) that could serve as indications of valid dietary reports. Further, we did not control for underreporters of energy intake (energy intake:basal metabolic rate < 1.35) due to the high number of underreporters in our data set (EP: 57%; LP: 69%). It is important to point out that underreporting is considerably high in overweight and obese pregnant women, in particular in late pregnancy (early pregnancy: 38%; late pregnancy: 49%) [26]. However, after post-hoc analysis, we reported no effect on the model after excluding underreporters. Nonetheless, despite having a large range of maternal protein intake, and considering the inclusion (or exclusion) or underreporters, we did not note any associations with offspring insulin sensitivity. This finding should be highlighted in particular because of the other potential associations between substantially low- and high protein intakes relative to recommendations during pregnancy and offspring health. For example, animal models have shown that protein restriction during pregnancy (rodent model) is linked to intrauterine growth restriction [27] and reduced offspring insulin signaling [28], and a high protein intake during pregnancy has been associated with increased offspring body weight and body fat [29]. Of note, protein restriction in studies using animal models is more severe (protein intake 8% of daily energy intake) compared to the average percent protein intake in the presented data (EP: 16.0% ± 3.4%; LP: 15.5% ± 3.2% of daily energy intake). Yet, there is a substantial range in the percent protein intakes at both time points (EP: 8.7%–25.4%; LP: 9.3%–28.4%) which is a strength of this cohort. Although likely difficult to perform, a randomized controlled trial of low compared to high dietary protein intake during pregnancy vs. offspring metabolic outcomes (e.g., insulin resistance) would provide more definite knowledge.

In addition to the amount of protein intake, the relationship between the type of dietary protein (e.g., animal, plant) and offspring measures is still unclear due to lack of studies and a disagreement between the available studies [3,4]. Recently-published work from our laboratory determined that there is a negative relationship between plant protein intake (and no relationship between animal protein intake) during pregnancy and maternal insulin resistance that disappeared when controlling for maternal BMI [1]. Therefore, to account for the impact of the type of protein consumed and also the amount of the type of protein consumed, in the current study we adjusted for plant protein intake quartiles during pregnancy at the respective time point (EP, LP) in addition to maternal BMI. Even after adjustments, no significant relationships were revealed, indicating that the type of protein likely does not exert any effect on the relationship between maternal total protein intake and offspring insulin resistance.

We had lower incidence of female births (42%) than would be expected by chance. After posthoc analyses we determined that there was no interaction effect of sex in the model at 12mo (β = −0.49, *p* = 0.40) and a trend toward significance at 24mo (β = 0.95, *p* = 0.06). Therefore, sex likely does not determine the outcome of the primary variable. Additionally, compared to other reports addressing similar variables (Maslova et al., 2014, *n* = 965 [4]; Maslova et al., 2017, *n* = 680 [3]; Shiell et al., 2000, *n* = 168 [5]), our sample size was relatively small (*n* = 182). However, the other studies were large-scale follow-ups of observational studies, and the present study was clinical in nature.

Nonetheless, several knowledge gaps in previous studies were improved in the current study. First, we present novel data of no association between maternal protein intake and insulin sensitivity in early postnatal years (1–2 years old), as compared to other studies finding mixed results in older offspring (9–40 years old) [3,4]. Nonetheless, long-term effects are important to study, and therefore, there are plans in place for the current cohort to be followed up until age eight years, and hopefully into adolescence, to determine trajectories and long term effects. In addition, the current study addressed maternal protein intake throughout pregnancy with recordings both at EP (8.4 ± 1.6 weeks) and LP (30.1 ± 0.4 weeks), but also averaged the dietary protein intake between EP and LP, whereas other studies were limited by a one-time assessment in late pregnancy (gestational week 25–30) [3,4,5]. Importantly, although we used the behavioral intervention that our group previously validated to reduce excessive GWG [14], just over 50% of the pregnant women in our cohort gained over the IOM recommendation for GWG. Indeed, we noted a decrease in protein intake relative to body mass from EP to LP (Table 2), indicating that it may have been difficult to consume the same amount of relative dietary protein throughout the duration of pregnancy as body mass increases.

## 5. Conclusions

Habitual self-reported dietary protein intake (amount or type) throughout pregnancy is not associated with indirect measurements of insulin sensitivity in offspring during the first two years of life. Maternal body mass index does not impact this relationship.

## Figures and Tables

**Table 1 nutrients-12-01338-t001:** Descriptive Characteristics, *n* = 182.

Characteristic	Value
Maternal Age (years)	29.7 ± 3.5
BMI, EP (kg/m^2^)	26.2 ± 4.2
BMI, LP (kg/m^2^)	29.4 ± 3.9 *
GWG (kg)	11.9 ± 4.3
Gestational Age (weeks)	39.4 ± 1.0
Delivery Mode (%)	
Vaginal	35.5
Vaginal Induced	29.0
C-Section	35.5
Offspring Sex, Female (%)	76 (42%)
Birth Weight (kg)	3.6 ± 0.5
Birth Length (cm)	51.2 ± 2.4
Feeding Mode (%)	
Breastfed	83
Formula fed	17
Length of Breastfeeding (months)	8.7 ± 6.4
Introduction of Solid Foods (months)	5.3 ± 1.6
Offspring HOMA2-IR, 12mo, *n* = 158	0.8 ± 0.8
Offspring HOMA2-IR, 24mo, *n* = 142	0.8 ± 0.9

Mean ± SD, * significantly different compared to EP, *p* < 0.05; BMI: body mass index; EP: early pregnancy at <10 weeks; LP: late pregnancy at 30 weeks; GWG: gestational weight gain; HOMA2-IR: homeostatic model assessment of insulin resistance; 12mo: 12 months; 24mo: 24 months.

**Table 2 nutrients-12-01338-t002:** Dietary Protein Intake during Early Pregnancy (EP), Late Pregnancy (LP), and Average Protein Intake throughout Pregnancy.

	EP *n* = 176	LP *n* = 176	Average *n* = 170
TEI (kcal/day)	1860.6 ± 449.0	1967.0 ± 472.5 ^‡^	1919.5 ± 399.3
TP (g/day)	74.0 ± 20.0	74.9 ± 19.2	74.4 ± 16.6
TP (g·kg^−1^·day^−1^)	1.1 ± 0.4	1.0 ± 0.3 *	1.0 ± 0.3
AP (g/day)	48.7 ± 17.5	48.4 ± 16.9	48.6 ± 13.9
AP (g·kg^−1^·day^−1^)	0.7 ± 0.3	0.6 ± 0.2 *	0.7 ± 0.2
PP (g/day)	25.4 ± 8.0	26.8 ± 7.4 ^‡^	26.0 ± 6.7
PP (g·kg^−1^·day^−1^)	0.4 ± 0.1	0.3 ± 0.1 ^‡^	0.4 ± 0.1

Mean ± SD, * significantly different compared to EP, *p* < 0.001; ^‡^ significantly different compared to EP, *p* < 0.05; EP: early pregnancy at <10 weeks; LP: late pregnancy at 30 weeks; TEI: total energy intake; TP: total dietary protein intake; AP: total dietary animal protein intake; PP: total dietary plant protein intake.

**Table 3 nutrients-12-01338-t003:** Regression Analyses Comparing Maternal Dietary Protein Intake at Early (EP) and Late (LP) Pregnancy and Offspring HOMA2-IR at 12 months (12mo) and 24 months (24mo).

	*n*	Protein at EP				*n*	Protein at LP			
		No Adj.		Adj.			No Adj.		Adj.	
		β	*p*	β	*p*		β	*p*	β	*p*
HOMA2-IR at 12mo	127	0.153	0.429	0.417	0.183	114	0.023	0.916	0.212	0.487
HOMA2-IR at 24mo	129	−0.349	0.098	−0.355	0.301	112	−0.442	0.085	−0.563	0.160

Mean ± Standard Deviation. **β** values represent the increase in HOMA2-IR for every 0.1 g·kg^−1^·day^−1^ increase in maternal dietary protein intake. EP: early pregnancy (<10 weeks); LP: late pregnancy (30 weeks); 12mo: 12 months; 24mo: 24 months. Adjustments were made for dietary plant protein intake quartiles and BMI during pregnancy at the respective gestational week (EP, LP), offspring sex, and offspring body fat percentage at the respective age (12mo, 24mo).

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
