# Peer review of "Dietary Protein Intake during Pregnancy Is Not Associated with Offspring Insulin Sensitivity during the First Two Years of Life"

_nutrients, 2020, doi:10.3390/nu12051338_

Round 1

Reviewer 1 Report

The present study examined associations between maternal protein intake and offspring insulin sensitivity (measured by HOMA) at 12 and 24 months, and did not observe any associations. While non-significant results are valuable, it is important to assess whether a study had sufficient statistical power to detect a biologically/ clinically important effect. Thus, the present study should include a power analysis.

The manuscript is generally clear and straightforward, but somewhat overstates the strengths of the present study.

One proposed strength of the current study was that it captured a wider range of protein intakes (e.g., line 229). However, it is not clear how/ why a wider ranger of protein intact was achieved, and while ranges of protein intake are described (line 233), it is not clear how many observations had high protein intakes (i.e., the wide range could have been caused by one or two observations, which would not be particularly powerful).  Moreover, to calculate the mass-corrected protein intakes for a previous study (lines 59-60), the present authors assumed a body mass of 80 kg. Is this realistic? The late gestation group in reference 10 had mean weight of 71.4 kg. Thus the range of mass-corrected protein intake in the previous study may have been higher. The range of protein intakes for the highest quartile in this previous study (ref 3) is lower than the calculated protein requirements in ref 10, is this realistic? I.e., not a single person in ref 3 met their protein requirements. It is not clear how adiposity would impact these protein requirements that are calculated based on total mass. If two individuals had the same lean mass, but one had a higher fat mass, I don’t see why the latter would have a higher protein requirement.

It is also suggested that a strength of the present study is that it examines early life (lines 85-87, 230, 320). However, whether this is a strength really depends on the question to be addressed. If the point is to investigate DOHAD effects, as suggested in the introductory paragraph, then it is more useful to examine longer-term effects.   

Adjusting for plant protein intake quartiles is a very strange adjustment to make. The analysis is testing whether HOMA2 is related to protein intake (i.e., animal + plant protein intake), controlling for plant protein intake quartile. Thus the intake quartiles are expected to be highly correlated with total protein intake. If the rationale is to examine the effects of animal protein intake independent of those of plant protein intake, then it would make more sense to include animal protein intake and plant protein intake as separate independent variables.

HOMA measures are often not normally distributed in which case they need to be logarithmically transformed (https://care.diabetesjournals.org/content/27/6/1487).

Minor

Slopes (beta) are described in the abstract, but these are only useful if the units are provided.

Line 28 Typo: 00711 (decimal missing)

Line 120. Interventions to “control for excessive gestational weight gain (GWG) during pregnancy and avoid excessive GWG” are described, but this issue is not addressed in the results. The mean GWG is 12 kg (Table 1), suggesting that many participants did show excessive GWG.

Line 125. Should be “aged”

Line 133. Should be “tared”

Line 148 A link should be provided for the HOMA2 calculator.

Table 1. Only 42% of offspring are female. Is this significantly different from expected, and if so, why might this be?

Line 174-176. If three hundred participants were recruited, why is the total cohort only 182?

Line 205. Offspring sex was included in analyses. It may be useful to also examine whether there is an interaction with sex, i.e., whether the effect of protein intake differs between the sexes.

A number of results are described in the Discussion section that are not mentioned in the Results section (lines 236, 269, 274, 284, 295).

Slopes and p-values for covariates (Table 3) should be included as supplementary material.

The paragraph in lines 251-268 is somewhat speculative and not directly relevant to the present work and so should be removed. This section discusses the maternal skeletal muscle, which was not assessed in the present study (it’s not clear which study the data described in lines 256-260 is from). The greater amount of fat free mass in obese women compared with normal weight women provides little insight into metabolic processes. They have greater muscle mass (and perhaps greater bone mass) because of greater loading, and the greater muscle mass does not offset the effects of increased fat. This paragraph actually undermines the importance of the present work (“rather than addressing the question of the impact of maternal diet on offspring insulin resistance, the more important question to address may be the relationship between offspring fat free mass and mitochondrial capacity”).

Line 305. Animal models of protein restriction probably use much more severe levels of protein restriction than those observed in the present study.

Author Response

Manuscript ID: nutrients-769290

Title: Dietary Protein Intake during Pregnancy is Not Associated with Offspring Insulin Sensitivity

Authors: Brittany R Allman, D. Keith Williams, Elisabet Børsheim *, Aline Andres *

Received: 25 March 2020

The authors would like to thank the reviewers for their time in the review process and for their constructive comments. Edits are addressed below:

REVIEWER 1

The present study examined associations between maternal protein intake and offspring insulin sensitivity (measured by HOMA) at 12 and 24 months, and did not observe any associations. While non-significant results are valuable, it is important to assess whether a study had sufficient statistical power to detect a biologically/ clinically important effect. Thus, the present study should include a power analysis. The manuscript is generally clear and straightforward, but somewhat overstates the strengths of the present study.

Of the original cohort (n = 300), we included every participant that satisfied the inclusion criteria for this sub-study (completed at least one of the two maternal visits and at least one of the two offspring visits, n = 182).

A multiple regression power analysis is done using a classic approach outlined in Maslova et al., 2017. We calculated the power to detect significance of a primary predictor term in the presence of 7 covariate terms, which matches our analysis. Our sample size had at least 80% power to detect a ‘small’ Cohen effect size of 0.05 at the 5% level of significance.  We include the power curves for Cohen effect sizes of 0.05 (small), 0.10, and 0.15 (medium).  We conclude that our study had sufficient power to detect a significant effect for protein if such an effect existed in our study population. We added the following statement on page 4:

Power analyses was estimated using a classic approach (Maslova et al., 2017) for multiple regression analyses in the presence of seven covariates. The current sample size (n = 182) had at least 80% power to detect a ‘small’ Cohen effect size of 0.05 at the significance level of 5%”.

  1. One proposed strength of the current study was that it captured a wider range of protein intakes (e.g., line 229). However, it is not clear how/ why a wider range of protein intact was achieved, and while ranges of protein intake are described (line 233), it is not clear how many observations had high protein intakes (i.e., the wide range could have been caused by one or two observations, which would not be particularly powerful).

The authors agree that defining how we captured a “wider range” of protein intakes is confusing. Further, after addressing the comment below (#2) and correcting the estimated body mass of pregnant women (gestational week 25) in Ref 3 to 67.0 kg (not 80 kg as originally in the manuscript, which is more realistic, we determined that our study did not actually have a wider spread of protein intakes compared to the two studies available (at least not at the higher range of protein intakes):

Current study*:         0.47-0.72 (Q1), 0.72-0.88 (Q2), 0.88-1.0 (Q3), 1.0-1.5 g·kg-1·day-1 (Q4)

Maslova 2017:          1.04-1.19 (Q1), 1.30-1.36 (Q2), 1.42-1.49 (Q3), 1.57-1.75 g·kg−1·d−1 (Q4)

Shiell:                         ≤0.96 (Q1), 0.97-1.13 (Q2), 1.13-1.29 (Q3), and >1.29 g·kg−1·d−1 (Q4)

*Average of EP and LP.

Therefore, the “wider range” of protein intake emphasis of the current manuscript has been removed from the Introduction and Discussion; however, the novelty remains that we analyzed offspring during early life (the first two years of life). We emphasize this aspect now throughout the manuscript.

  1. Moreover, to calculate the mass-corrected protein intakes for a previous study (lines 59-60), the present authors assumed a body mass of 80 kg. Is this realistic? The late gestation group in reference 10 had mean weight of 71.4 kg. Thus the range of mass-corrected protein intake in the previous study may have been higher. The range of protein intakes for the highest quartile in this previous study (ref 3) is lower than the calculated protein requirements in ref 10, is this realistic? I.e., not a single person in ref 3 met their protein requirements.

Please see the response to Comment #1. We have edited the introduction section to reflect these changes. The estimated protein intake for Ref 3 has been recalculated using the revised estimated body mass, and added to the discussion on page 7:

Our report of no relationship between a relatively wide range of maternal protein intake (LP range: 0.4 - 1.8 g·kg-1·day-1) and offspring insulin resistance measures is confirmed by another report in pregnant women in gestational week 25 that reported no relationship between maternal protein intake (1.04-1.75 g·kg−1·d−1, calculated from estimated body mass) and fasting plasma insulin or HOMA-IR in offspring aged 9-16 years old [3]. Considering other reports of a positive association between protein intake and insulin resistance measures (reviewed in Rietman et al., 2014 [20]), it is interesting to note that the protein intake in both studies spanned higher than protein requirements recently reported in late pregnancy (1.55 g·kg−1·d−1 [21]), yet neither were associated with offspring insulin resistance.

  1. It is not clear how adiposity would impact these protein requirements that are calculated based on total mass. If two individuals had the same lean mass, but one had a higher fat mass, I don’t see why the latter would have a higher protein requirement.

Protein requirements, in general, have been published per unit of body mass (in kg), regardless of body composition. Dietary protein requirements during the difference stages of pregnancy have been determined (early pregnancy: 11 – 20 weeks, 1.2 g·kg−1·d−1; late pregnancy: 30 – 38 weeks, 1.52 g·kg−1·d−1) in healthy women using the indicator amino acid oxidation method and measurement of the oxidation rate of L-[1-13C]phenylalanine to 13CO2 (Stephens TV et al., Protein Requirements of Healthy Pregnant Women during Early and Late Gestation Are Higher than Current Recommendations. J. Nutr. 2015, 145, 73–78). This is a whole-body method that does not discern between lean body mass and fat mass, and therefore, the requirements calculated as a result of this method are true requirements per kg body mass in that particular study population. We agree that further studies should be done to determine the impact of lean body mass and fat mass on requirements, as it seems plausible that the lean body mass would be the strongest predictor/driver for the protein requirements.

  1. It is also suggested that a strength of the present study is that it examines early life (lines 85-87, 230, 320). However, whether this is a strength really depends on the question to be addressed. If the point is to investigate DOHAD effects, as suggested in the introductory paragraph, then it is more useful to examine longer-term effects.

DOHAD effects begin as early as fetal development, and perhaps even as far back as oocyte development, as our group has previously published (Ruebel ML et al., Obesity Modulates Inflammation and Lipid Metabolism Oocyte Gene Expression: A Single-Cell Transcriptome Perspective. J Clin Endo Met. 2017, 102(6) 2029-2038). Our question encompasses the first two years of life, which would therefore fall under that umbrella and has not been reported previously. However, we do agree that long-term effects are important to study. The cohort is being followed up until age 8 years at the present and we hope to continue the follow-up into adolescence. Once the data are available for these later ages, we will continue our investigation and provide updates on long term effects. The following was added on page 8:

Nonetheless, long-term effects are important to study, and therefore, there are plans in place for the current cohort to be followed up until age eight years, and hopefully into adolescence, to determine trajectories and long term effects.

  1. Adjusting for plant protein intake quartiles is a very strange adjustment to make. The analysis is testing whether HOMA2 is related to protein intake (i.e., animal + plant protein intake), controlling for plant protein intake quartile. Thus the intake quartiles are expected to be highly correlated with total protein intake. If the rationale is to examine the effects of animal protein intake independent of those of plant protein intake, then it would make more sense to include animal protein intake and plant protein intake as separate independent variables.

Although we agree that this would be another prudent statistical approach to take, we chose our current approach for a few key reasons. First, our previous report (Allman BR et al., Obesity Status Affects the Relationship between Protein Intake and Insulin Sensitivity in Late Pregnancy. Nutrients. 2019 11(9)) determined that maternal dietary plant protein (and not animal protein) intake is positively related to measures of maternal insulin sensitivity in late pregnancy. Therefore, we wanted our focus to be on the potential influence of maternal dietary plant protein intake on offspring insulin sensitivity measures.

Then we chose to compute the quartiles not only for use as a covariate but also as it added a descriptive element to our study (i.e., define the quartiles for the percentage of plant protein for our study subjects; note that the percentage of animal protein would be complement to the plant protein). Nonetheless, as a response to your comment, we have run the same analyses using the percentage plant protein intake (as a continuous variable), instead of plant protein quartiles. We found that the results were no different between adjusting for the percentage plant protein and plant protein quartiles (table below). Therefore, we have decided to keep our statistical approach consistent with our original approach (quartiles vs. percentage).

12mo

24mo

Plant Protein Percentage

β = 0.691, P = 0.075

β = -0.302, P = 0.383

Plant Protein Quartiles

β = 0.687, P = 0.078

β = -0.270, P = 0.440

Furthermore, in our original model (with adjustment of plant protein quartiles), we tested for multi-collinearity by calculating the Variance Inflation Factor [VIF], which provides a VIF factor for each of the predictors. Because the maximum VIF of each of the predictors was not greater than 10 at either of the time points (12mo: 2.1; 24mo: 2.2), we conclude that we have no influence of multi-collinearity with our samples.

  1. HOMA measures are often not normally distributed in which case they need to be logarithmically transformed (https://care.diabetesjournals.org/content/27/6/1487).

We have logarithmically transformed our HOMA measures, and found that there were still no notable relationships between average maternal protein intake and offspring HOMA-β at either time point (12mo: β = 0.47, p = 0.08; 24mo: β = 0.03, p = 0.85), or HOMA2-IR at either time point (12mo: β = 6.883e-01, p = 0.06; 24mo: β = -0.63, p = 0.11). There results have been added to the results on page 5:

After logarithmic transformation of HOMA2-IR and HOMA-β, there were still no notable relationships between average maternal protein intake and offspring HOMA2-IR at either time point (12mo: β = 6.883e-01, p = 0.06; 24mo: β = -0.63, p = 0.11), or HOMA-β at either time point (12mo: β = 0.47, p = 0.08; 24mo: β = 0.03, p = 0.85).

However, because: 1) the conclusions using the original statistical approach and the logarithmic approach are identical, and 2) we want to be able to compare the HOMA2-IR units between our previous manuscript (maternal protein intake vs. maternal HOMA-IR), other papers, and this follow-up manuscript, we have decided to focus on the original model (maternal protein vs. offspring HOMA2-IR, without HOMA logarithmic transformation and HOMA- β).

Minor

  1. Slopes (beta) are described in the abstract, but these are only useful if the units are provided.

HOMA2-IR has no units, and therefore are not provided when expressing slopes (beta). However, the unit of protein intake used (g·kg−1·d−1) was provided earlier in the sentence.

  1. Line 28 Typo: 00711 (decimal missing)

Corrected.

  1. Line 120. Interventions to “control for excessive gestational weight gain (GWG) during pregnancy and avoid excessive GWG” are described, but this issue is not addressed in the results. The mean GWG is 12 kg (Table 1), suggesting that many participants did show excessive GWG.

The approximate percentage of women that gained over the IOM recommendations for GWG has been added to the results section (and in the Discussion):

Importantly, although we used the behavioral intervention that our group previously validated to reduce excessive GWG [14], just over 50% of the pregnant women in our cohort gained over the IOM recommendation for GWG (not pictured in Table 1).

  1. Line 125. Should be “aged”

Corrected.

  1. Line 133. Should be “tared”

Corrected.

  1. Line 148 A link should be provided for the HOMA2 calculator.

This has been added.

  1. Table 1. Only 42% of offspring are female. Is this significantly different from expected, and if so, why might this be?

We do recognize that we had a lower incidence of female births compared to male births. This has been addressed in the limitations section:

We had lower incidence of female births (42%) than would be expected by chance. After posthoc analyses we determined that there was no interaction effect of sex in the model at 12mo (β = -0.49, p = 0.40) and a trend toward significance at 24mo (β = 0.95, p = 0.06). Therefore, sex likely does not determine the outcome of the primary variable.

  1. Line 174-176. If three hundred participants were recruited, why is the total cohort only 182?

Although this rationale is explained in the sentence immediately after, we have added an additional sentence to more clearly explain why only part of the original cohort could be used:

Many participants from the parent study were excluded for not satisfying the requirements for inclusion in the current analyses (i.e., a maternal dietary record from at least one of the two pregnancy time points; successful offspring blood draw during at least one of the two offspring time points).

  1. Line 205. Offspring sex was included in analyses. It may be useful to also examine whether there is an interaction with sex, i.e., whether the effect of protein intake differs between the sexes.

We agree and we have added some verbiage to the limitation section to address this point. There was no interaction effect of sex. Please see the answer to number 7 above.

  1. A number of results are described in the Discussion section that are not mentioned in the Results section (lines 236, 269, 274, 284, 295).

Each of these posthoc analyses are now added to the statistical approach in the methods section (2.4.3 Posthoc Analyses of Quartile Analysis), as well as the results section (3.4 Results of Posthoc Analyses of Quartile Analysis).

  1. Slopes and p-values for covariates (Table 3) should be included as supplementary material.

Supplementary tables (Supplementary Table 1 and 2) have been created to present the slopes and p-values for covariates within Table 3, respectively.

  1. The paragraph in lines 251-268 is somewhat speculative and not directly relevant to the present work and so should be removed. This section discusses the maternal skeletal muscle, which was not assessed in the present study (it’s not clear which study the data described in lines 256-260 is from). The greater amount of fat free mass in obese women compared with normal weight women provides little insight into metabolic processes. They have greater muscle mass (and perhaps greater bone mass) because of greater loading, and the greater muscle mass does not offset the effects of increased fat. This paragraph actually undermines the importance of the present work (“rather than addressing the question of the impact of maternal diet on offspring insulin resistance, the more important question to address may be the relationship between offspring fat free mass and mitochondrial capacity”).

The authors agree that this paragraph is speculative and detracts from the importance of the present work, and it has therefore been removed.

  1. Line 305. Animal models of protein restriction probably use much more severe levels of protein restriction than those observed in the present study.

We agree that protein restriction studies in an animal model are particularly restrictive (most provide protein at ~8% of daily energy intake). Nonetheless, although the average percent protein intake at both time points was higher than in protein restricted animal studies (EP: 16.0 ± 3.4%; LP: 15.5 ± 3.2% of daily energy intake), our data do indeed show that there is a substantial range at both time points (EP: 8.7-25.4%; LP: 9.3-28.4%). Additionally, we believe that the women in the present study were acclimated to their habitual diet (a strength of observational studies), compared to animal studies with protein restriction whereby animals are randomized into a dietary condition to which they are not accustomed. We do agree, however, that it would be important (however, likely difficult) to design a randomized control trial of low compared to high dietary protein intake during pregnancy and investigate its effect on offspring metabolic outcomes (e.g., insulin resistance). The following sentence has been added:

Of note, protein restriction in studies using animal models is more severe (protein intake 8% of daily energy intake) compared to the average percent protein intake in the presented data (EP: 16.0 ± 3.4%; LP: 15.5 ± 3.2% of daily energy intake). Yet, there is a substantial range in the percent protein intakes at both time points (EP: 8.7-25.4%; LP: 9.3-28.4%) which is a strength of this cohort.

Reviewer 2 Report

This manuscript investigated the relationship between maternal dietary protein intake and insulin sensitivity in offspring during the first two years of life. And the Authors found no relationship.

The manuscript is a bit confusing, the continuous repetition of data in parentheses makes reading difficult, especially in the discussion. The results should be expanded with all the new information contained in the discussion.

My comments are:

  • Was the remaining of the diet investigated in terms of macronutrients?
  • Were there vegetarians or vegans in the group of participating women? if yes, were there differences in protein intake?
  • Introduction: the paragraph from line 49 through line 67 it is too verbose and belongs to the discussion;
  • Table 2: how do the Authors explain the fact that total dietary protein intake decreases during pregnancy?
  • Discussion: the sentences from 255 through 260; 269 through 278; 283 through288; 298 through 300; 324 through 327, belong to the results.

Author Response

Manuscript ID: nutrients-769290

Title: Dietary Protein Intake during Pregnancy is Not Associated with Offspring Insulin Sensitivity

Authors: Brittany R Allman, D. Keith Williams, Elisabet Børsheim *, Aline Andres *

Received: 25 March 2020

The authors would like to thank the reviewers for their time in the review process and for their constructive comments. Edits are addressed below:

REVIEWER 2

This manuscript investigated the relationship between maternal dietary protein intake and insulin sensitivity in offspring during the first two years of life. And the Authors found no relationship. The manuscript is a bit confusing, the continuous repetition of data in parentheses makes reading difficult, especially in the discussion. The results should be expanded with all the new information contained in the discussion. My comments are:

  1. Was the remaining of the diet investigated in terms of macronutrients?

Being a follow-up up our previously-published study comparing maternal dietary protein intake vs. maternal insulin resistance (Allman BR et al., Obesity Status Affects the Relationship between Protein Intake and Insulin Sensitivity in Late Pregnancy. Nutrients. 2019 11(9)), the main focus of the present manuscript was maternal dietary protein intake (not the other macronutrients) vs. offspring insulin resistance. However, our plan is to analyze the full spectrum of macronutrients compared to a multitude of additional offspring outcomes in another paper.

  1. Were there vegetarians or vegans in the group of participating women? If yes, were there differences in protein intake?

There were no vegetarians or vegans in our population of pregnant women. This has been added on page 5 in the manuscript.

  1. Introduction: the paragraph from line 49 through line 67 it is too verbose and belongs to the discussion;

The authors agree that this paragraph was quite long, therefore, it has been truncated.

  1. Table 2: how do the Authors explain the fact that total dietary protein intake decreases during pregnancy?

The last sentence of the discussion considered the decrease in protein relative to body mass from EP to LP, which explains the decrease in total dietary protein intake during pregnancy. The absolute intake doesn’t decrease, rather the adjusted value for body weight does:

Indeed, we noted a decrease in protein intake relative to body mass from EP to LP (EP: 1.09 ± 0.36; LP: 0.97 ± 0.28 g·kg-1·day-1, p = 0.0003), indicating that it may have been difficult to consume the same amount of relative dietary protein throughout the duration of pregnancy as body mass increases.

  1. Discussion: the sentences from 255 through 260; 269 through 278; 283 through288; 298 through 300; 324 through 327, belong to the results.

We agree that these posthoc analyses should be moved to the results, and therefore, the results section has been expanded to include them.

Reviewer 3 Report

Authors studied protein intake during pregnancy, and evaluate the relationship between protein intake and offspring HOMA2-IR. They showed that EP protein, LP protein and average protein did not associate with offspring HOMA2-IR at 12mo or 24mo.

This paper is controversial to previous papers, because the subjects were in infancy and have not been followed until adulthood.

The HOMA2-IR used in the method seems unreasonable for application in infancy. The number of subjects is small. Large number cohort study is required.

There are a number of issues that need to be addressed:

Major:

  • No significant difference was due to the narrow range of protein intake in the diet (~12% of energy to ~19% of energy). It is thought that significant differences occur in animal experiments due to considerable differences in protein intake.
  • Because the subject are infants, the title should also include infants.
  • There are reports that the intestinal flora changes with spontaneous delivery or cesarean section. Subject should indicate spontaneous delivery or cesarean section.
  • In infants, nutritional status could affect insulin resistance. The subjects should indicate whether they were breast feeding or artificial nutrition.
  • The insulin clamp method is the golden standard for the evaluation of insulin resistance. HOMA2-IR is a simple method and has problems in evaluation especially for infants.
  • In some subjects, fasting an infant for 15 or 17 hours is ethically problematic.
  • Was there any difference in insulin secretory ability, such as HOMA-beta, other than insulin resistance?
  • Is there evidence that even infants can make a difference in muscle quality?

Minor:

  • Table 2.

Why is animal protein intake lower than in previous reports?

  • Table 3.

HOMA2-IR real figures and raw data should be displayed.

Author Response

Manuscript ID: nutrients-769290

Title: Dietary Protein Intake during Pregnancy is Not Associated with Offspring Insulin Sensitivity

Authors: Brittany R Allman, D. Keith Williams, Elisabet Børsheim *, Aline Andres *

Received: 25 March 2020

The authors would like to thank the reviewers for their time in the review process and for their constructive comments. Edits are addressed below:

REVIEWER 3

Authors studied protein intake during pregnancy, and evaluate the relationship between protein intake and offspring HOMA2-IR. They showed that EP protein, LP protein and average protein did not associate with offspring HOMA2-IR at 12mo or 24mo. This paper is controversial to previous papers, because the subjects were in infancy and have not been followed until adulthood. The HOMA2-IR used in the method seems unreasonable for application in infancy. The number of subjects is small. Large number cohort study is required. There are a number of issues that need to be addressed:

Major:

  1. No significant difference was due to the narrow range of protein intake in the diet (~12% of energy to ~19% of energy). It is thought that significant differences occur in animal experiments due to considerable differences in protein intake.

Our data showed that there was a substantial range in the percentage of daily energy intake contributed by protein at both time points (EP: 8.7-25.4%; LP: 9.3-28.4%). And further, it seems that there was a relatively even spread of the percentage of protein from daily energy intake, evidenced by how close the average intakes and median intakes were at both EP (mean: 16.0%; median: 15.8%) and LP (mean: 15.5%; median: 15.1%) time points. Additionally, women in the present study were acclimated to their habitual diet (a strength of observational studies), compared to animal studies with protein restriction (typically ~8% of daily energy intake) whereby animals are randomized into a dietary condition to which they are not accustomed (although they have time to adjust). We do agree, however, that it would be important (however, likely difficult) to design a randomized controlled trial of low compared to high dietary protein intake during pregnancy vs. offspring metabolic outcomes (e.g., insulin resistance). We have added the following to page 8:

Although likely difficult to perform, a randomized controlled trial of low compared to high dietary protein intake during pregnancy vs. offspring metabolic outcomes (e.g., insulin resistance) would provide more definite knowledge.

  1. Because the subject are infants, the title should also include infants.

We have changed the title to include information about the population:

Dietary Protein Intake during Pregnancy is Not Associated with Offspring Insulin Sensitivity during the First Two Years of Life

  1. There are reports that the intestinal flora changes with spontaneous delivery or cesarean section. Subject should indicate spontaneous delivery or cesarean section.

The percentage of vaginal (35.5%), vaginal induced (29.0%), and C-section (35.5%) deliveries have been added to Table 1.

  1. In infants, nutritional status could affect insulin resistance. The subjects should indicate whether they were breast feeding or artificial nutrition.

The mode of feeding (e.g., breast feeding, formula fed) and the length of breastfeeding have been added to Table 1.

  1. The insulin clamp method is the golden standard for the evaluation of insulin resistance. HOMA2-IR is a simple method and has problems in evaluation especially for infants.

We do understand that the gold standard for evaluation of insulin resistance evaluation is the clamp technique. However, this technique poses serious ethical concerns in toddlers and has never been performed in this population. The only instance where it can be ethically defended to perform this technique in this population is when babies are being treated in an intensive clinical setting (e.g., preterm infants) (Sarlengo KM et al., Relationship between glucose utilization rate and glucose concentration in preterm infants. Biol Neonate, 1986, 49(4):181-9).

Additionally, we do agree that HOMA2-IR poses potential problems regarding the use of it to determine insulin resistance in our population. However, the measurement has been used in a pediatric population age 2 years old (e.g., Nithun T et al., Association of Acanthosis Nigricans and Insulin Resistance in Indian Children and Youth - A HOMA2-IR Based Cross-Sectional Study. Indian Dermatol Online J, 2019, 10(3):272-278). In our manuscript we have outlined the limitations, but have also explained that HOMA standards, per se, have not yet been defined in a toddler population. Furthermore, although other options to measure insulin resistance (e.g., glucose tests, modeling) have been used in childhood populations, toddlers are an understudied population. Further, because our large cohort study drew one-time fasted blood samples at 12mo and 24mo, we were limited by what we could ultimately use. Therefore, we decided to use HOMA2-IR because it is the updated version of HOMA, accounting for variations in hepatic and peripheral glucose resistance.

Nonetheless, taking into account the range of HOMA2-IR values in our data, we also explain that posthoc analysis using a robust regression model (using median, and not mean) revealed that the outlier HOMA2-IR values did not influence the original model. Each of these points have been addressed in the discussion:

Although the hyperinsulinemic-euglycemic clamp is considered the gold standard for insulin resistance assessment [21], it poses serious ethical concerns in a toddler population (e.g., IV infusion, long commitment to laboratory) and has only been used in infant populations simultaneously undergoing critical care (e.g., preterm infants [22]). Furthermore, although other options to measure insulin resistance (e.g., glucose tests, modeling) have been used in older pediatric populations, toddlers are an understudied population [23]. Further, because our large cohort study drew one-time fasted blood samples at 12mo and 24mo, we were limited by which insulin resistance assessment we could use. Because: 1) HOMA-IR was used in other work comparing the same variables [3]; 2) HOMA2-IR is a computerized updated model of HOMA-IR accounting for variations in hepatic and peripheral glucose resistance; 3) it has been used in a pediatric population [24]; 4) our previous publication that serves as the precursor of this study also used HOMA2-IR [1]; and 5) the measurement takes both fasting glucose and insulin into account, we decided to use HOMA2-IR as an assessment of insulin resistance. Nevertheless, there was a significant spread in HOMA2-IR of the offspring at both 12mo (0.06-6.85) and 24mo (0.07-5.26). Importantly, the large variation between offspring fasting duration before the blood draw (12mo: 0-15 hours; 24mo: 0-17 hours) did not affect the model and thus the conclusions, as revealed in our posthoc analysis. However, because the prevalence and the specific definition of insulin resistance using HOMA2-IR in toddlers (12-24 mo) has not been defined, it may be difficult to discover a potential relationship at this age. Therefore, future research should aim to define insulin resistance in the early years. Nonetheless, our posthoc analysis using a robust regression utilizing the median (instead of the mean, to determine if outliers were driving the model) determined that there was no change in the model, and therefore, an effect of particularly high HOMA2-IR values on our findings is likely non-existent.

  1. In some subjects, fasting an infant for 15 or 17 hours is ethically problematic.

Our ethical considerations for our participants are paramount. We asked mothers to visit the laboratory when their child was approximately 4-hours fasted at the 12mo time point (average ± STD: 2 ± 2 hours; range: 0-15 hours), and approximately 12-hours fasted overnight at the 24mo time point (average ± STD: 6 ± 6 hours; range: 0-17 hours). At the time of their visit to the laboratory, the mother was asked about the time when the child last ate. Even so, we determined that the wide range in fasting times had no effect on the original model after posthoc analysis (Section 3.4 Results of Posthoc Analyses of Quartile Analysis). The section within the methods has been adjusted to define our original fasting criteria:

Mothers were asked to visit the laboratory when their child was approximately 4-hours fasted at the 12mo time point and approximately 12-hours fasted overnight at the 24mo time point. At the time of their visit to the laboratory, the mother was asked about the time when the child last ate. Blood samples were collected from the offspring after 2 ± 2 (average±SD) hours of fasting at 12mo (range: 0-15 hours), and 9 ± 6 hours of fasting at 24mo (range: 0-17 hours).

  1. Was there any difference in insulin secretory ability, such as HOMA-beta, other than insulin resistance?

We have conducted analyses using HOMA-β as the primary outcome variable using the same model. The conclusion was that there were no relationships with average maternal protein intake and HOMA-β (12mo: β = 57.6, p = 0.08; 12mo: β = 1.29, p = 0.94). This has been added to the results section on page 5.

  1. Is there evidence that even infants can make a difference in muscle quality?

Because the discussion on muscle quality was quite speculative and not directly relevant to the present work, we removed that section.

Minor:

  1. Table 2. Why is animal protein intake lower than in previous reports?

To our knowledge, the animal protein intake in our study population is comparable to others. For example, Maslova et al., (2014) reported that overall, pregnant women consumed approximately 0.6 – 1.3 g·kg-1·day-1. On average, we reported approximately 0.6 and 0.7 g·kg-1·day-1 at EP and LP time points, respectively.

  1. Table 3. HOMA2-IR real figures and raw data should be displayed.

The actual figures related to these analyses were underwhelming and do not provide any additional information that Table 3 does not provide, and therefore we decided to not add them to the manuscript. Offspring HOMA2-IR raw data is displayed in Table 1, and maternal dietary protein intake raw data is displayed in Table 2. We defer to the editor for guidance on this comment.

Round 2

Reviewer 2 Report

I am satisfied with the changes made

Reviewer 3 Report

Authors responded to the reviewers’ questions properly.

The revised version is improved compared with the original one.